# Role of Efflux Pumps on Antimicrobial Resistance in *Pseudomonas aeruginosa*

**DOI:** 10.3390/ijms232415779

**Published:** 2022-12-13

**Authors:** Andre Bittencourt Lorusso, João Antônio Carrara, Carolina Deuttner Neumann Barroso, Felipe Francisco Tuon, Helisson Faoro

**Affiliations:** 1Laboratory for Applied Science and Technology in Health, Carlos Chagas Institute, Fiocruz, Curitiba 81350-010, Brazil; 2School of Medicine and Life Sciences, Pontifícia Universidade Católica do Paraná, Curitiba 80215-901, Brazil; 3Laboratory of Clinical Veterinary Parasitology, Federal University of Paraná, Curitiba 80035-050, Brazil; 4Laboratory of Emerging Infectious Diseases, Pontifícia Universidade Católica do Paraná, Curitiba 80215-901, Brazil; 5CHU de Quebec Research Center, Department of Microbiology, Infectious Disease and Immunology, University Laval, Quebec, QC G1V 0A6, Canada

**Keywords:** efflux pumps, antimicrobial resistance, *Pseudomonas aeruginosa*

## Abstract

Antimicrobial resistance is an old and silent pandemic. Resistant organisms emerge in parallel with new antibiotics, leading to a major global public health crisis over time. Antibiotic resistance may be due to different mechanisms and against different classes of drugs. These mechanisms are usually found in the same organism, giving rise to multidrug-resistant (MDR) and extensively drug-resistant (XDR) bacteria. One resistance mechanism that is closely associated with the emergence of MDR and XDR bacteria is the efflux of drugs since the same pump can transport different classes of drugs. In Gram-negative bacteria, efflux pumps are present in two configurations: a transmembrane protein anchored in the inner membrane and a complex formed by three proteins. The tripartite complex has a transmembrane protein present in the inner membrane, a periplasmic protein, and a porin associated with the outer membrane. In *Pseudomonas aeruginosa*, one of the main pathogens associated with respiratory tract infections, four main sets of efflux pumps have been associated with antibiotic resistance: MexAB-OprM, MexXY, MexCD-OprJ, and MexEF-OprN. In this review, the function, structure, and regulation of these efflux pumps in *P. aeruginosa* and their actions as resistance mechanisms are discussed. Finally, a brief discussion on the potential of efflux pumps in *P. aeruginosa* as a target for new drugs is presented.

## 1. The Silent Pandemic of Antimicrobial-Resistant Bacteria

Antibiotics revolutionized medicine in the 1930s–1940s, allowing millions of lives to be saved due to their ability to end infections caused by pathogenic bacteria. The first antibiotic used to treat an infectious disease was arsphenamine, also known by the trade name Salvarsan. This synthetic molecule, discovered by Paul Ehrlich, was widely used for the treatment of syphilis [1]. In time, the first commercially available broad-spectrum antibiotic was penicillin, discovered by Ian Flemming in 1929 [2]. Penicillin was introduced and widely used in the 1940s, during World War II, and helped to save many soldiers suffering from war wounds. However, its indiscriminate use quickly led to the emergence of the first penicillin-resistant bacterial strains [3].

Antimicrobial resistance (AMR) is an old pandemic that has become a major public health crisis. According to the World Health Organization (WHO), the topic is considered one of the most important threats to the human species in the 21st century [4]. It is a natural phenomenon shaped by natural selection and evolution. However, in recent decades there has been an acceleration in the emergence of AMR bacteria due to selective pressures caused by excessive or inappropriate use of antibiotics [5]. A relationship between antibiotic consumption, incorrect prescriptions, and the spread of resistant strains of bacteria has already been reported, as well as a high rate (30–50%) of incorrect prescriptions of these drugs [6]. In addition, there is a lack of interest on the part of the largest pharmaceutical companies in developing new antibiotics due to the high cost and low financial return, resulting in a significant innovation gap in the development of antibiotics [7]. Incorrect prescriptions, extensive antibiotic use in agriculture and livestock, and the low availability of new antimicrobials are some of the reasons for the worsening antibiotic resistance crisis. A global study that evaluated the impact of AMR on public health showed that, in 2019, most deaths in the world resulted directly from or were associated with resistant bacteria, which were a contributing factor in the loss of 4.95 million lives [8]. To put this in perspective, estimates of the number of deaths caused by the SARS-CoV-2 virus in the first year of the pandemic (2020) range between 1.8 and 3 million [9]. The effects of the SARS-CoV-2 pandemic slowed down due to mass vaccination and, to a large extent, socio-educational measures. In the long term, it is possible to envisage that the same type of measures will be applied to the use of antibiotics. Unfortunately, the AMR pandemic is not slowing down. In 2017, WHO published a list containing the priority antibiotic-resistant pathogens for the development of new drugs, which considered bacteria with resistance to multiple drugs to be of critical importance and to pose a threat to hospitals and patients who require devices such as ventilators and blood catheters [10]. The number of species of bacteria capable of causing infections is quite large; however, a group of bacteria called the ESKAPE pathogens stands out due to its high prevalence in nosocomial infections [11]. The ESKAPE group is formed by the bacteria *Enterococcus faecium*, *Staphylococcus aureus*, *Klebsiella pneumoniae*, *Acinetobacter baumannii*, *Pseudomonas aeruginosa*, and species of the genus *Enterobacter*.

Bacteria are known to employ a range of different resistance mechanisms. According to CARD database [12], the antimicrobial resistance mechanisms can be classified as: (i) the production of enzymes that modify the antibiotic molecule, leading to inactivation or destruction of the drug; (ii) changes in membrane permeability, which prevents the absorption of external substances; (iii) protection or alteration of target; (iv) target superexpression; (v) target replacement; and, (vi) efflux pumps that expel toxic components [13,14]. Over the years, the selective pressure caused by the continuous use of various antimicrobial agents has led microorganisms to accumulate various resistance mechanisms. Whether by horizontal transfer of resistance genes or selection of resistant mutants, the acquisition of several resistance factors has led to the emergence of multidrug-resistant (MDR) bacteria.

## 2. Pseudomonas Aeruginosa

*P. aeruginosa* is a Gram-negative bacillus commonly found in many environments, notably in soil and water exposed to intense human activity (e.g., wastewater-, hydrocarbon-, and pesticide-contaminated soil) [15]. It is an opportunistic bacteria related to healthcare infections, including ventilator-associated pneumonia (VAP), intensive care unit infections, central line-related bloodstream infections, surgical site infections, urinary tract infections, burn wound infections, keratitis, and otitis media [16]. This pathogen affects immunocompromised patients, in part due to its ability to evade both innate and acquired immune defenses through adhesion, colonization, and biofilm formation, and to produce various virulence factors that cause significant tissue damage. It also causes diseases with a high mortality rate in patients diagnosed with cystic fibrosis, neonatal infections, cancer, and severe burns [17,18,19]. Patients with cystic fibrosis present a higher incidence of colonization and infection by *P. aeruginosa*. Epithelial cells of the lung can ingest the invading *P. aeruginosa*, followed by desquamation, thus, protecting lungs from infection. However, patients with cystic fibrosis phagocytose less *P. aeruginosa* [20]. This reduced phagocytosis is associated with internal system defect and protection of specific *P. aeruginosa* LPS present in the external membrane of the bacteria [21].

*P. aeruginosa* frequently displays both multidrug resistance (MDR), when the bacterium is resistant to at least one agent in three or more antimicrobial categories, and extensive drug resistance (XDR), when the bacteria remain susceptible to only one or two categories [22,23,24]. According to data from the CDC, in 2017, 33,600 cases of infection and 2700 deaths from *P. aeruginosa* MDR were identified, associated with an expenditure of USD 767 million [25]. A strong aggravating factor in an infection by *P. aeruginosa* is its ability to produce biofilms. A biofilm is a structure formed by a bacterial community aggregated by components of the extracellular matrix secreted by the cells that compose it. The biofilm presents a protective environment for cells against abiotic stresses, immune system, and antibiotic action, which makes it difficult to eliminate the infection [16]. 

This bacterium was isolated for the first time in 1882 from cutaneous wounds of two patients with bluish–green pus, reported by Carle Gessard [26]. After that, several reports associated *P. aeruginosa* as the cause of blue–green purulence in patients’ wounds. The characteristic pigment, called pyocyanin, caught the researchers’ attention years before the bacterium was isolated. It was extracted by Fordos in 1860 and was associated with rod-shaped organisms in 1862 [27]. *P. aeruginosa* grows on most culture media used in routine analysis, and it can grow at 42 °C, a characteristic that distinguishes it from other *Pseudomonas* species. Additionally, colonies are often green due to the production of pyocyanin. Pyocyanin is an important virulence factor in infections caused by *P. aeruginosa* due to its toxic effects, putting cellular systems under increased oxidative stress [28]. Moreover, high concentrations of pyocyanin are present in the sputum of patients with chronic pulmonary disorders resulting from bacteremia [29]. Rudolf Emmerich and Oscar Löw showed in the 1890s that green bacteria (*P. aeruginosa*) isolated from patient bandages inhibited the growth of other bacteria [27,30]. Later, it was discovered that this antibiotic action was due to an enzyme called pyocyanase, which is considered to be the first clinic antibiotic used to treat human infections in hospitals [27].

The *P. aeruginosa* reference strain for genetics and functional analyses of physiology and metabolism is PAO1. *P. aeruginosa* PAO1 is a spontaneous chloramphenicol-resistant mutant of the original PAO strain that was isolated in 1954 from a wound in Melbourne, Australia [31,32]. A genetic map of its chromosome was constructed using the transduction and conjugation mechanisms of gene exchange in bacteria [33]. A physical map of the PAO1 genome was created using pulsed-field gel electrophoresis (PFGE) and afterward combined with the genetic map data [34,35]. The PAO1 strain was fully sequenced by 2000, and the genome is relatively large (5.5–7 Mbp) compared with other sequenced bacteria [36]. This indicates that the *P. aeruginosa* genome encodes a large proportion of regulatory enzymes important for metabolism, transportation, and efflux of organic compounds, which contributes to its intrinsic resistance to antibiotics and high adaptability to environmental changes. The contribution of efflux pumps to the emergence of MDR bacteria is significant. This is directly related to the ability of an efflux pump to export different classes of drugs [37]. Multidrug resistance has been demonstrated in all ESKAPE group bacteria, including *P. aeruginosa* [11].

## 3. Function and Structure of Efflux Pumps

Six families of efflux pumps have been described in the literature as having multidrug efflux capacity (Table 1): the small multidrug resistance (SMR) family; the major facilitator superfamily (MFS); the resistance/nodulation/cell division (RND) family; the ATP-binding cassette (ABC) superfamily; the multidrug and toxic compound extrusion (MATE) family [38,39], and the recently described proteobacterial antimicrobial compound efflux (PACE) family [40]. Active efflux systems may be responsible for resistance to several chemically distinct antibiotics and bactericides, with alarming numbers of occurrences in environmental and clinical isolates [41,42,43]. Such systems have demonstrated a practically ubiquitous presence in all kingdoms [44,45], sharing mechanisms [39,46], and helping to increase the dynamics of resistance profiles [47,48].

Protein transport systems received much attention in the 1990s, with structural, functional, and phylogenetic studies that aimed to uncover their distinct evolutionary origins [39,43,44,49,50]. These first studies, focused on the RND (super)family, enabled the identification of three common protein units, an inner transmembrane transporter, a periplasmic protein, and an outer membrane channel protein, which combine to form a trans-periplasmic channel [45,51,52] that allows the capture of substrates in the periplasm or cytoplasm [45]. After the assembly of the protein complex, the system stabilizes in two different stages: a resting stage, and a transport stage [53,54]. Driven by the proton motive force, the RND substrate efflux goes through cycles of site access, substrate binding, and extrusion, all mediated by a strictly coordinated rotational mechanism [55] that allows the system to work as a pump mechanism. As other families began to be discovered and more effort started to be put in the study of efflux pump mechanisms, different structures and driving forces began to be described, such as single protein mechanisms [56,57,58,59,60], ion gradients [56,57], and ATP [59] dependent pumps.

Generally, the genes that encode efflux pumps are chromosomal, with a smaller number present on plasmids; the proteins have conserved motifs containing glycine, proline, aspartate, and other hydrophobic residues in addition to tandem-repeat structures linked to secondary structures and coiled-coil and β-strands [51,61] (reviewed elsewhere [39]). Taken together, these characteristics distinguish the class from other outer membrane proteins, helping the complex to function as a barrier to prevent drug access to its target [42,51,62].

**Table 1 ijms-23-15779-t001:** Major families of bacterial efflux pumps in *P. aeruginosa*.

Family ^1^	Efflux Pump	Gene Ids ^2^	Substrates ^3^	References
ABC	Ttg2 (Mla)	PA4456-PA4455-PA4454-PA4453-PA4452	CHL, CIP, COL, DMF, DOX, LVX, MIN, OFX, TET, TGC, TOB, TMP	[63,64,65]
	PA1874-77	PA1874-PA1875-PA1876-PA1877	CIP, GEN, NOR, TOB	[66,67]
	PA3228	PA3228	CAR, LVX, NOR	[68]
MFS	Mfs1	PA1262	PQT	[69]
	Mfs2/SmvA	PA1282	OCT, PQT	[69,70]
	CmlA1	GNT62_RS22140	CHL	[71,72]
SMR	PASmr/EmrE_Pae_	PA4990	ACR, EtBr, GEN, KAN, NEO	[73,74]
	SugE subfamily SMR	PA1882	Further research is needed	[75]
MATE	PmpM	PA1361	ACR, BZK, CIP, EtBr, NOR, OFX, TPPCL	[76]
PACE	PA2880	PA2880	CHX	[77]
RND	MexAB-OprM	PA0425-PA0426-PA0427	AMI, AMX, ATM, CAR, CR, MA, FEP, CFP, CFSL, CTX, FOX, CZOP, CPO, CES, CAZ, CZX, CRO, CXM, CHL, CTET, CIN, CIP, CLX, DP, DOR, ENX, ERY, FMOX, GEN, IPM, LVX, CLM, MEM, MOX, NAF, NAL, NOR, NOV, OFX, OMC, OTC, PG, PPA, PIP, PTZ, PMA, SPX, SPI, STN, SUL, TZB, TET, TIC, TOS	[78,79,80,81]
	MexCD-OprJ	PA4599-PA4598-PA4597	AMX, MA, FEP, CFP, CFSL, CTX, FOX, CZOP, CPO, CES, CZX, CRO, CXM, CHL, CHX, CTET, CIN, CIP, CLX, DOR, ENX, ERY, FMOX, LVX, CLM, MEM, NAF, NAL, NOR, NOV, OFX, OMC, OTC, PG, PPA, PIP, PMA, SPX, SPI, TET, TOS	[79,80,82,83]
	TMexCD-TOprJ	LSG45_RS29735-LSG45_RS29740-LSG45_RS29745	FEP, CEQ, CAZ, CTET, CIP, DOX, ERV, FLO, GEN, MIN, NAL, OTC, STR, TET, TGC	[84,85]
	MexEF-OprN	PA2493-PA2494-PA2495	CHL, QN, TET, TMP	[86]
	MexGHI-OpmD	PA4205-PA4206-PA4207-PA4208	5-Me-PCA, ACR, EtBr, NOR, R6G, TET, V	[87,88,89]
	MexJK-OprM	PA3677-PA3676-PA0427	ERY, TET, TCS	[90]
	MexMN-OprM	PA1435-PA1436-PA0427	BAL30072, ATM, BIPM, CAR, CMN, CAZ, CFT, CHL, MEM, MET, MOX, NOV, PIP, SUL, TMC, TP, TIC	[91,92]
	MexPQ-OpmE	PA3523-PA3522-PA3521	Hoechst 33342, CHL, CIP, ERY, KIT, NOR, RKM, TET, TPPCL	[91]
	MexVW-OprM	PA4374-PA4375-PA0427	ACR, CPO, CHL, ERY, EtBr, NOR, OFX, TET	[93]
	MexXY-OprM (-OprA)	PA2019-PA2018-PA0427 (-PSPA7_3271)	ACR, AMI, AMX, MA, FEP, CFP, CFSL, CTX, FOX, CZOP, CPO, CAZ, CZX, CRO, CXM, CHL, CTET, CIN, CIP, CLX, DOR, ENX, ERY, EtBr, FMOX, GEN, IPM, KAN, LVX, CLM, MEM, NAF, NAL, NEO, NOR, OFX, OMC, OTC, PG, PPA, PIP, PTZ, PMA, SPX, SPI, STR, TZB, TET, TIC, TOB, TOS, CAR (only with OprA), SUL (only with OprA)	[79,80,94,95,96]
	MuxABC-OpmB	PA2528-PA2527-PA2526-PA2525	ATM, ERY, KIT, NOV, RKM, TET	[97]
	TriABC-OpmH	PA0156-PA0157-PA0158-PA4974	TCS	[98]

^1^: ABC: ATP-binding cassette transporters; MFS: major facilitator superfamily; SMR: small multidrug resistance; MATE: multidrug and toxic-compound extrusion; PACE: proteobacterial antimicrobial compound efflux; RND: resistance-nodulation-cell division. ^2^: IDs obtained from the *Pseudomonas* Genome Database [99]. ^3^: ACR: acriflavine; AMI: amikacin; AMX: amoxicillin; ATM: aztreonam; BIPM: biapenem; BZK: benzalkonium chloride; CAR: carbenicillin; CAZ: ceftazidime; CEQ: cefquinome; CES: cefsulodin; CFP: cefoperazone; CFSL: cefoselis; CFT: ceftolozane; CHL: chloramphenicol; CHX: chlorhexidine; CIN: cinoxacin; CIP: ciprofloxacin; CLX: cloxacillin; CMN: carumonam; COL: colistin; CPO: cefpirome; CR: carvacrol; CRO: ceftriaxone; CTET: chlortetracycline; CTX: cefotaxime; CXM: cefuroxime; CZOP: cefozopran; CZX: ceftizoxime; DMF: dimethylformamide; DOR: doripenem; DOX: doxycycline; DP: dipyridyl; ENX: enoxacin; ERV: eravacycline; ERY: erythromycin; EtBr: ethidium bromide; FEP: cefepime; FLO: florfenicol; FMOX: flomoxef; FOX: cefoxitin; GEN: gentamicin; IPM: imipenem; KAN: kanamycin; KIT: kitasamycin; LCM: lincomycin; LVX: levofloxacin; MA: cefamandole; MEM: meropenem; MET: methicillin; MIN: minocycline; MOX: moxalactam; NAF: nafcillin; NAL: nalidixic acid; NEO: neomycin; NOR: norfloxacin; NOV: novobiocin; OCT: octenidine; OFX: ofloxacin; OMC: oleandomycin; OTC: oxytetracycline; PG: penicillin G; PIP: piperacillin; PMA: piromidic acid; PPA: pipemidic acid; PQT: paraquat; PTZ: piperacillin-tazobactam; QN: quinolones; R6G: rhodamine 6G; RKM: rokitamycin; SPI: spiramycin; SPX: sparfloxacin; STN: streptonigrin; STR: streptomycin; SUL: sulbenicillin; TCS: triclosan; TET: tetracycline; TGC: tigecycline; TIC: ticarcillin; TMC: temocillin; TMP: trimethoprim; TOB: tobramycin; TOS: tosufloxacin; TP: thiamphenicol; TPPCL: tetraphenylphosphonium chloride; TZB: tazobactam; V: vanadium.

### 3.1. General Role of Efflux Pumps in Bacteria in a Natural Environment

Efflux pumps evolved as a way for bacteria to interact with the environment [100]. Their ubiquitous presence in nature [101] allowed bacteria to survive in their ecological niches [38], protecting them from toxic compounds produced by other species of bacteria or by the host, antimicrobial molecules, reactive oxygen species (ROS), and toxic byproducts of biochemical degradation pathways [38,43,52,102,103,104]. Efflux pumps can be specific for only one substrate or can export a range of molecules [38,62,105]. This difference in substrate selection is directly associated with the physicochemical characteristics of the substrate and the molecular interactions between the substrate, protein binding site, and environment, possibly reflecting the ability of the efflux pumps to capture their substrates [106]. As such, some of the main substrates of efflux pumps cited in the literature are detergents [52,62,107,108], antibiotics [52,62,102,108,109], organic solvents [102,110], dyes [62,102,107,108,109], disinfectants, aromatic hydrocarbons [109], cationic biocides [111], free fatty acids [112], lipophilic drugs, amphiphilic agents [107], antiseptics [62,108], anticancer drugs, uncouplers [108], and anti-virulence compounds [43]. 

The transcriptional regulation of efflux systems depends directly on the concentration of external signals [113], together with the action of internal transcription factors such as the TetR family proteins [114], LysR-type transcriptional regulators (LTTRs) [115], the multidrug regulator OstR [113], and activators MarA, Rob, and SoxS [112]. Cross-talking between efflux systems [113] and the need for chaperone-assisted folding increases its complexity, as observed in the downregulation of OmpF porin, which reduces cell permeability [52]. As observed, the presence of efflux pumps contributes to the maintenance of cell viability, virulence, and quorum sensing [43,102], assisting the bacterial cell in tolerating rapidly changing environments [101]. In this case, mechanisms of virulence and resistance evolved to fulfill different functions in the natural environment earlier than and not directly related to their role in multidrug resistance [43].

### 3.2. Efflux Pumps Involved in AMR in Bacteria

The first study that directly evaluated the action of efflux pumps on antibiotic resistance dates to the 1980s and revealed the action of plasmid-transferred energy-dependent efflux pumps on tetracycline resistance in *Escherichia coli* [46]. McMurry and colleagues demonstrated the ability of efflux pumps to shift the drug even against a concentration gradient [46], showing a strict relationship between the efflux pump together with outer membrane permeability and the acquisition of drug resistance [42,52].

Efflux systems have been found in comparable numbers among pathogenic and nonpathogenic bacteria. Its emergence earlier than the extensive exposure to antimicrobial agents suggested that their evolution as a resistance mechanism occurred independently of the use of antimicrobials [44]. The efflux system allows the bacteria to survive at low levels of drugs, conferring time to develop a more specialized mechanism of resistance [116]. When combined with the outer membrane barrier, the efflux system guarantees cellular protection from a range of compounds [106], favoring the dissemination of gene coding for such systems. Some isolates have genes for different efflux pumps in their genome, acting as genetic reservoirs [48,52]. This organization can also occur from gene duplication events [117] and gene cassettes [116]. 

Efflux pump specificity can follow a nonlinear behavior depending on the variety of compounds it can transport, leading to efficient extrusion even in cases of low substrate affinity for the binding site [106]. At the same time, such bonding can lead to the blocking of rotational movement and impediment of extrusion [53]. In this way, the structural changes observed can allow the binding of two or more substrate molecules in the same system, such as the extrusion of hydrophobic and hydrophilic compounds, expanding the specificity of the efflux pump [55].

The role of efflux pumps in AMR can be characterized as a byproduct of physiological functions that are beneficial to bacteria [38], and their efficiency depends on several factors, both internal and external to the bacterium. The presence of efflux pump genes can be considered the first step in the acquisition of resistance to antibiotics [118].

## 4. Efflux Pumps as a Mechanism of Antimicrobial Resistance in *P. aeruginosa*

*P. aeruginosa* has a naturally low susceptibility to antibiotics, explained by its extensive intrinsic resistome [119], low membrane permeability [120], and ability to form biofilms [16]. In its core genome, there are several antibiotic resistance genes, such as *bla*_ampC_, and the genes encoding the multidrug efflux pumps (Mex) MexXY, MexAB-OprM, MexCD-OprJ, and MexEF-OprN [121], all of which are part of the RND superfamily.

These efflux pumps are tripartite protein complexes, which are drug/proton antiporters that catalyze the extrusion of their specific substrates from the periplasm through the outer membrane [122]. They are composed of three different proteins: a periplasmic adaptor protein, commonly known as periplasmic membrane fusion protein (PMFP), such as MexA, MexX, MexC, or MexE; a resistance-nodulation-cell division transporter (RNDt), such as MexB, MexY, MexD, or MexF, and a channel-forming outer membrane factor (OMF), such as OprM, OprJ, or OprN [123]. Structural analyses of the MexAB-OprM efflux pump show that the PMFP forms a stable complex associated with the inner membrane and the RNDt, both of which are recruited by the OMF, which resides in the outer membrane [124,125] (Figure 1).

As discussed before, efflux pumps are major factors in developing multidrug resistance in bacteria. The four multidrug efflux pumps in *P. aeruginosa* are involved in the extrusion of toxic molecules and contribute to reduced antibiotic susceptibility [126]. Furthermore, overexpression of these multidrug efflux pumps in *P. aeruginosa* is directly associated with resistance to most anti-pseudomonal drugs [80,127] and can potentially reduce the efficiency of new classes of drugs developed against *P. aeruginosa* and other Gram-negative pathogens, such as LpxC inhibitors [128,129] and cefiderocol [130]. These results highlight the importance of taking efflux mechanisms into consideration when developing antipseudomonal drugs.

Other efflux pumps identified in *P. aeruginosa*, such as MexJK and MexGHI-OpmD, have been mainly associated with quorum-sensing modulation and cell-to-cell communication [88,131] and will not be discussed further in this review.

### 4.1. MexAB-OprM

The operon MexAB-OprM was the first multidrug efflux pump reported in *P. aeruginosa* [78] and is considered the main contributor to antibiotic resistance, with MexA as the PMFP, MexB as the RNDt, and OprM as the OMF [132]. When the drug concentration increases near the pump, MexB goes through a conformational change and can eject active molecules toward a tunnel formed by MexA and OprM across the periplasm and outer membrane [133].

MexAB-OprM is constitutively expressed in wild-type strains, and its expression is controlled mainly by the repressor genes *mexR* [134], *nalC* [135], and *nalD* [136]. The use of antibiotics selects *P. aeruginosa* strains with increased MexAB-OprM expression [137,138]. Several mutations in repressor genes have been associated with this phenomenon [139], in particular mutations causing translational disruption, such as nonsense substitutions and frameshifts [138,140], disruption by insertion sequences [139,141], and non-synonymous substitutions that alter the molecular structure of the repressors [137,142]. However, there are commonly reported non-synonymous substitutions, such as V126E in *mexR*, which are not associated with an increase in efflux pump expression [143,144].

MexAB-OprM can export several antibiotics, including quinolones, macrolides, tetracyclines, lincomycin, chloramphenicol, novobiocin, and most β-lactams [79], and it has been shown to be a major factor in the resistance to the herbal antimicrobial compound carvacrol [81]. Overexpression of this efflux pump is associated with resistance to most antipseudomonal antibiotics, except for colistin [80], and carbapenemase-producing carbapenem-resistant *P. aeruginosa* strains often show an increased *mexAB-oprM* expression, which could be contributing to its resistance to carbapenems [145]. Moreover, it has been shown that overexpression of MexAB-OprM and AmpC, a chromosomally encoded class C β-lactamase that contributes to the resistance to many penicillins [146,147], has synergistic effects on the resistance of *P. aeruginosa* to most antipseudomonal β-lactams, except for ceftolozane/tazobactam, imipenem, and imipenem/relebactam [148].

### 4.2. MexXY

The MexXY efflux pump is formed by the PMFP MexX and the RNDt MexY and is the only multidrug efflux pump operon in *P. aeruginosa* that does not contain a coding sequence for an outer membrane factor. However, it does form a multidrug efflux pump together with OprM from the MexAB-OprM operon [94]. In some strains, such as *P. aeruginosa* PA7 [149], the MexXY operon contains a coding sequence for another outer membrane factor called OprA, which can also form a complex with MexXY [95,150].

Its expression is antimicrobial-inducible and is controlled mainly by the repressors MexZ [151], the two-component regulatory system ParRS [152], and by the aminoglycoside-inducible anti-repressor ArmZ [153,154]. Several inactivating or single amino acid mutations in *mexZ* have been associated with an increased expression of MexXY [144,155,156], particularly in cystic fibrosis isolates [125,157], a phenomenon that has also been reported for *parR* [125,152,158] and *parS* [125,152].

MexXY is known to contribute to aminoglycoside resistance [159], and its overexpression is commonly observed in strains bearing aminoglycoside-modifying enzymes (AMEs), in which MexXY and the AME work in synergy to promote aminoglycoside resistance [156]. Moreover, it has been shown that MexXY is also associated with resistance to most antipseudomonal antibiotics, similarly to MexAB [80], and it is particularly important in cystic fibrosis lung isolates of *P. aeruginosa* that harbor a defective MexAB pump [160]. In the OprA-harboring strain *P. aeruginosa* PA7, MexXY is also able to expel and confer resistance to carbenicillin and sulbenicillin, but only when partnering with the OMF OprA [95]. Moreover, specific single amino acid substitutions in MexY have been linked to increased resistance to aminoglycosides, cefepime, and fluoroquinolones, highlighting the importance of the MexXY system for antibiotic resistance in *P. aeruginosa* [160].

### 4.3. MexCD-OprJ

The MexCD-OprJ operon encodes an efflux pump that is usually silent or expressed at a low level in *P. aeruginosa* [161], which normally does not contribute to this pathogen’s natural resistance to antibiotics; however, it is associated with resistance to several classes of antibiotics when overexpressed [82,162]. Its expression is mainly regulated by the repressor *nfxB*, which is transcribed divergently from the *mexCD-oprJ* operon [82]. Several mutations in *nfxB* have been associated with an increased expression of MexCD-OprJ, such as nucleotide deletions [163], missense, and nonsense mutations [162].

MexCD-OprJ is mainly associated with resistance to fluoroquinolones such as levofloxacin and ciprofloxacin [80,161], but can also extrude other antimicrobial agents such as macrolides, novobiocin [79], tetracyclines, chloramphenicol [162,164,165], zwitterionic cephalosporins, such as cefepime and cefpirome [166], and the biocide chlorhexidine [83]. Additionally, mutations in *mexD* may change the substrate specificity of the efflux pump and have been associated with resistance to carbenicillin [167], ceftolozane-tazobactam, and ceftazidime-avibactam [163]. However, strains that overexpress MexCD-OprJ commonly show greater susceptibility to aminoglycosides and some β-lactams [166,168,169], while the overexpression of this efflux pump may lead to an increased susceptibility to complement-mediated killing [170]. Recently, Sanz-García and colleagues showed that exposure to subinhibitory concentrations of ciprofloxacin led to a selection of mutants with increased expression of MexCD and cross-resistance to other antibiotics [100].

Interestingly, variations of *mexCD-oprJ* genes have been found in mobile genetic elements in *P. aeruginosa* and several other bacteria [84,171]. A group of these mobile efflux pump operons have been called *tmexCD-toprJ* (“t” for transferable), and data suggest that they originated from the chromosome of a *Pseudomonas* spp. [85]. Several variations of the original *tmexCD-toprJ* have been identified since, either chromosomally or plasmid located, and have been directly associated with resistance to tetracyclines and reduced susceptibility to several other classes of antibiotics under laboratory conditions [84,85].

### 4.4. MexEF-OprN

As for MexCD-OprJ, the MexEF-OprN system is generally inactive under normal conditions but has been associated with resistance to chloramphenicol, quinolones, and trimethoprim when overexpressed [86]. The pump is regulated by MexT, a LysR-like transcriptional activator [172], and MexS, a putative oxidoreductase of unknown function [172,173], both encoded by genes located upstream of the *mexEF-oprN* operon (Figure 1). Interestingly, *mexT* is commonly found with inactivating mutations in wild-type *P. aeruginosa* strains, and overexpression of MexEF-OprN arises from the reversion of these mutations [174]. The *mexS* gene expression is also activated by MexT [172], and several mutations in *mexS*, commonly observed in clinical strains, are known to cause overexpression of MexEF-OprN [173,175,176,177].

Several other genes have also been associated with the modulation of *mexEF-oprN* expression. Iftikhar et al. [178] reported that a *P. aeruginosa* mutant with a defective *pvcB* gene from the *pvc* paerucumarin synthesis operon showed significant repression in mRNA levels of *mexEF-oprN*, *mexT,* and *mexS*, suggesting the involvement of the *pvc* operon in the transcriptional regulation of *mexEF-oprN*. It has also been shown that *mvaT* and *ampR*, global expression regulators in *P. aeruginosa*, negatively regulate MexEF-OprN [179,180], and data suggest the existence of still unknown MexEF-OprN regulators [177].

Overexpression of MexEF-OprN is also associated with decreased expression of the porin OprD because of the MexEF-OprN activator MexT acting as an oprD repressor [172,181]. Inactivating mutations and downregulation of oprD have been associated with resistance to carbapenems [139,182] and colistin [183], which explains why strains overexpressing MexEF-OprN are commonly resistant to antibiotics not extruded by this efflux pump.

## 5. Efflux Pumps and Biofilm Formation

Bacterial biofilms consist of microbial communities structured and organized in a matrix of exopolysaccharides (EPS), produced and secreted by the organisms themselves, adhered to a surface [184]. Efflux pumps play an important role in bacterial biofilm formation. Most studies in this area show that efflux pump expression is upregulated in biofilms, conferring greater antibiotic resistance [16]. There is an important positive feedback relationship between biofilms and efflux pumps. Tests of efflux pump inhibitors showed a significant reduction in biofilm in different bacteria [185]. The mechanism that underpins the relationship between efflux pumps and biofilm formation differs between bacterial species, but three main mechanisms have been identified [186]. Firstly, biofilm formation requires several molecules to be transported to the extracellular environment, a function that depends partially on efflux pumps [187]. Secondly, intercellular communication for biofilm formation is called quorum sensing, a type of molecule-mediated communication that requires certain molecules to be released into the extracellular environment. Without these molecules, biofilms cannot obtain the structure of a mature biofilm, expand, or release planktonic cells [188]. Thirdly, some antibiotics act on the regulation and synthesis of elements for the biofilm; therefore, efflux pumps that expel antibiotics are related to the maintenance of the biofilm [189]. 

Acyl-homoserine lactones (AHL) are important molecules for biofilm production in *P. aeruginosa*, as they facilitate quorum sensing [190]. Pearson et al. showed that the inhibition of efflux pumps led to an increase in the intracellular concentration of AHL and a reduction in the extracellular concentration, suggesting a clear association between efflux pumps and elements in biofilm formation [191]. The MexAB-OprM pump plays an important role in *P. aeruginosa* quorum sensing by delivering AHLs to the extracellular environment [187]. Furthermore, the overexpression of efflux pumps on *P. aeruginosa* can improve biofilm formation, with effects including denser structure, as demonstrated by Sánchez et al. [192]. *P. aeruginosa* with mutations on *nalB* and *nfxB* overexpresses MexAB-OprM and MexCD-OprJ, respectively, resulting in a denser biofilm formation compared with wild-type *P. aeruginosa*. 

The MexEF-OprN efflux pump drives the efflux of 4-hydroxy-2-heptylquinoline (*Pseudomonas* quinolone signal), which is used by *P. aeruginosa* to facilitate quorum sensing [193]. Interestingly, studies comparing the transcriptome of planktonic and sessile cells (basal cells of the biofilm) demonstrate an upregulation of several genes associated with efflux pumps such as *mexAB-oprM* and *mexCD-oprJ* [36,194,195]. The work by Harrington et al. evaluated the *P. aeruginosa* transcriptome in an ex vivo pig lung model and a sputum model, showing a repression in MexGHI-OmpD expression in the lung model-associated biofilm [196]. Thus, it is possible to conclude that pump efflux interferes with biofilms but also that biofilms interfere with efflux pump regulation.

## 6. Efflux Pumps as Targets for New Drugs

Since the role of efflux pumps as a mechanism of resistance to several antimicrobials became known, significant resources have been devoted to developing molecules that can inhibit this system. Studies in this area began by attempting to find antibiotic-like molecules expelled from the bacteria to inhibit these pumps, such as derivatives of tetracyclines. Despite several in vitro studies, these drugs have never been clinically tested, mostly for reasons of serum instability and toxicity [197]. The first efflux pump inhibitors studied were verapamil, 2,4-dinitrophenol, lansoprazole, protonophore carbonyl cyanide m-chlorophenylhydrazone, and Phe-Arg-ß-naphthylamide. The latter is a dipeptide, a class of molecules that has been studied for their efflux pump inhibitory properties [198,199,200,201,202]. In *P. aeruginosa*, Phe-Arg-ß-naphthylamide increases the activity of quinolones, decreasing their expulsion from the bacterial cell and lowering their minimum inhibitory concentrations [203]. After these, several other similar molecules were studied, such as MC-207,110, MC-04,112, MC-02,434, MC-510,051, MC-04,124, and MC-02,595 [204]. MC-04,124 showed similar results to the other compounds in the series; however, the authors considered it to be a less toxic drug [205,206]. Peptides have also been successfully employed to inhibit SMR family efflux pumps. Mitchell and collaborators synthesized several peptides that targeted the TM4 region of *P. aeruginosa* SMR. The TM4 domain is important for homodimer assembly and for a functional SMR pump. They showed that the synthesized peptides were able not only to decrease the efflux via SMR but also to improve the biocidal activity of other compounds [207]. In addition to synthetic components, dozens of natural and herbal products have been tested for use as efflux pump inhibitors; while some have shown inhibitory activity, clinical studies have not yet been carried out [208].

Revisiting existing drugs with safe clinical use is an alternative to the development of new molecules. In vitro studies have shown that mefloquine, a well-known molecule widely used in the treatment of malaria [209], acts as an efflux pump inhibitor in *P. aeruginosa* [210]. Trimethoprim, a drug used in combination with sulfamethoxazole to treat various infections, also has efflux pump inhibitory activity. The use of trimethoprim together with ciprofloxacin has a synergistic effect, increasing sensitivity in *P. aeruginosa* to ciprofloxacin, although the mechanism is still unclear [211]. A further study with trimethoprim and sertraline, using *Galleria mellonella* as an in vivo infection model, showed the effectiveness of both molecules as synergists in levofloxacin treatment compared with levofloxacin alone. Significant increases in *P. aeruginosa*-infected *G. mellonella* survival were seen with treatment with levofloxacin and trimethoprim, or levofloxacin and sertraline, compared with monotherapies [212]. On the other hand, Laborda and collaborators tested a library of 1243 natural products for molecules capable of both increasing the expression of efflux pumps and providing them with a substrate without leading to an increase in antibiotic resistance in *P. aeruginosa*. The authors hypothesized that increasing the expression of these genes and the efflux of the inducing molecule could overcome the efflux of antibiotics while decreasing the efflux of virulent molecules to the outside. They found four compounds that were able to increase the expression of the *mexCD-oprJ* or *mexAB-oprM* genes while also decreasing the virulence of *P. aeruginosa* in tests with *Caenorhabditis elegans*: two coumarin-like compounds, one 1,4-benzodioxan-like compound, and one 4-chloroindole compound [213]. Additionally, using a drug library selection strategy, Tambat and collaborators found that the ethyl 4-bromopyrrole-2-carboxylate molecule can act synergistically with antibiotics, lowering their minimum inhibitory concentration in strains that overexpress RND family transporters [214]. Finally, comparative and structural studies between MexB and MexY from *P. aeruginosa* have shown that the presence of bulky tryptophan residues in the hydrophobic pit prevents binding of the efflux pump inhibitor ABI-PP [215].

Despite several potential candidates for efflux pump inhibitors in *P. aeruginosa*, obtaining an effective drug is still the subject of several studies. *P. aeruginosa* constitutively expresses several efflux pumps with different substrate specificities, different physicochemical properties and different efflux constants (K_E_). Understanding these variables and how the compounds correlate with these different penetration mechanisms is crucial to effectively combat *P. aeruginosa* antibiotic resistance. It is also interesting to report that these efflux pump inhibitors can also affect biofilm production since there is a close relationship between the efflux of signaling molecules for biofilm formation and quorum sensing, as discussed earlier [216]. The action on quorum sensing molecules has been shown in transcriptome studies, where Phe-Arg-β-naphthylamide was used to reduce the expression of genes related to biofilms [217].

## 7. Concluding Remarks

*P. aeruginosa* is an opportunistic pathogen of extreme clinical importance that is responsible for most secondary respiratory tract infections in patients with cystic fibrosis or undergoing intubation. The emergence of MDR and XDR strains of *P. aeruginosa* is a serious aggravating factor to this problem. Efflux pumps are among the most important AMR mechanisms that lead to the emergence of MDR and XDR strains. These protein complexes evolved from a system of interaction with other bacteria, the environment, and hosts with a survival resource. Efflux pumps can export dozens of different compounds into the extracellular environment, such as quorum sensing molecules, virulence factors, and toxic compounds such as antibiotics. This primary mechanism of antibiotic resistance is responsible for reducing the intracellular concentration of the drug to a sub-inhibitory concentration at which other more robust and specific resistance mechanisms can evolve and be selected for. As such, efflux pumps in *P. aeruginosa* have received significant attention in relation to the development of new drugs. Although several genes have been associated with the transcriptional regulation of efflux pump genes, *P. aeruginosa* has a complex transcriptional regulatory network in which many genes can be affected by unique global expression modulators. In addition, putative transcription factors with no known targets have been identified through in silico approaches. Additional studies are needed to accurately identify and characterize all trans-acting elements and substrates involved in efflux pump regulation, which may be crucial for the design of new antipseudomonal drugs. Unfortunately, the big pharmaceutical companies have abandoned investments in the development of new drugs due to the high associated cost. In this scenario, it may be necessary to establish public–private partnerships between governments, pharmaceutical companies and the WHO, aiming at the discovery or development of new antibiotic molecules.

## Figures and Tables

**Figure 1 ijms-23-15779-f001:**
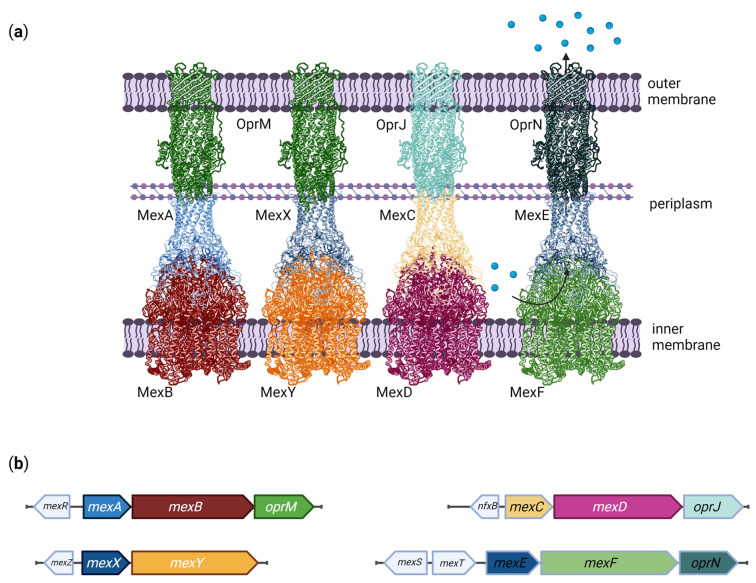
Multidrug efflux pumps in *P. aeruginosa*. (**a**) Schematic structures of the four main efflux pumps involved in antibiotic resistance in *P. aeruginosa*, showing the resistance-nodulation-cell division transporters (MexB, MexY, MexD and MexF) on the inner membrane; the periplasmic membrane fusion proteins (MexA, MexX, MexC and MexE) on the periplasm; and the channel-forming outer membrane factors (OprM, OprJ and OprN) on the outer membrane. Protein representations based on the Protein Databank (PDB) structures: 2V50, 4DK1 and 3D5K. (**b**) Organization of the Mex operons and upstream regulatory genes in the *P. aeruginosa* genome. Regulatory genes are represented in white, and the Mex coding sequences follow the same color patterns of the protein structures shown in (**a**). Created with BioRender.com.

## Data Availability

Not applicable.

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
