# Peer review of "Role of Efflux Pumps on Antimicrobial Resistance in Pseudomonas aeruginosa"

_ijms, 2022, doi:10.3390/ijms232415779_

Round 1

Reviewer 1 Report

The authors have written an interesting review on the area of efflux pumps in Pseudomonas aeruginosa. While not a particularly novel subject for a review article, timely updates in important fields of research such as this are always welcome. See the attached file for my specific comments on the manuscript - once these have been addressed, I think the article will be suitable for publication. To this end, I would like to commend the authors for their hard work in putting together this review.

Author Response

REVIEWER #1

Corrections for authors to make:

Lines 18-20: long sentence! Consider breaking it up into smaller, simpler sentences.

HF: Thanks for the suggestion. The text has been corrected accordingly.

Lines 20-22: incorrect! You need to mention here that i) you are talking exclusively about Gram-negative bacteria, since Gram-positive bacteria don’t have a periplasm or outer membrane, and ii) only certain (super)families of efflux pumps are known to form these tripartite assemblies (RND, MFS, ABC).

HF: We appreciate the note and have corrected the text accordingly.

Line 23: Latin names of microorganisms should always be italicized. Please correct this throughout the manuscript (there are numerous instances to correct)

HF: The names have been corrected

.

Lines 25-27: avoid using personal pronouns.

HF: The text has been modified.

Line 31: awkward sentence – please revise

HF: The sentence has been modified to improve understanding.

Line 33: this sentence is, at very best, misleading. Please look up arsphenamine and prontosil, then revise paragraph accordingly.

HF:  We apologize for the inconsistency. The text has been corrected accordingly.

Line 40: too many commas

HF: The text has been modified.

Line 55: the COVID-19 pandemic didn’t just ‘slow down over time’, it was brought under control through a range of societal-level control measures! It is absolutely plausible that the implementation of control measures for antibiotic use on this scale (e.g. global ban on dispensing antibiotics without a prescription, global limitations of use of antibiotics in livestock etc) could have a similarly positive effect on the AMR pandemic. This could be highlighted briefly.

HF: The text was modified to highlight the importance of socio-educational measures in combating the SARS-CoV-2 and AMR pandemic.

Line 60: ‘...however, a group of bacteria called the ESKAPE pathogens stands out...’

HF: Corrected

Line 62: the word ‘bacteria’ should not be italicized.

HF: Corrected

Line 64: ‘Bacteria are known to employ a range of different resistance mechanisms, such as...’

HF:  The text has been modified.

Line 68: additionally, there are target-mimicking resistance mechanisms (see qnr based fluoroquinolone resistance)

HF: Thanks for the suggestion. We list in this description the resistance mechanisms described in the CARD database, according to which the qnr system is classified as "target protection" (https://card.mcmaster.ca/ontology/36558). We include this information in the text to improve understanding.

Lines 68-71: some comment on how this happens? Horizontal vs vertical evolution?

HF: We have included an additional comment.

Lines 78-79: delete ‘extensive drug resistance’

HF: Corrected

Lines 94-95: this is probably incorrect. While pyocyanase may be recognised as the first antibiotic used in a hospital setting, there is evidence that things like copper were used to treat wounds as far back as ancient Egypt.

HF: Thanks for the suggestion. Indeed, there have been other methods of treating infections, including numerous herbal preparations. However, pyocyanase is in a different position from these other ancient/alternative therapies. The importance of pyocyanase was recognized by Dr. Selman Waksman in his 1942 article:

"The active agent has been isolated, purified, and crystallized only in very few instances. Pyocyanase was the first antagonistic substance to have thus been obtained (Emmerich and Low, 1899). Several others have been isolated recently."

Waksman SA, Woodruff HB. Selective Antibiotic Action of Various Substances of Microbial Origin. J Bacteriol. 1942 Sep;44(3):373-84. doi: 10.1128/jb.44.3.373-384.1942. PMID: 16560575; PMCID: PMC373686.

Finally, we modified the sentence to include highlighting the fact of the clinical use of pyocyanase in hospitals.

Line 117: what is meant here by ‘AMR factors’? It isn’t clear

HF: Thank you for pointing that out. The sentence has been removed.

Line 126: ‘Gram’ should always be capitalized. Change throughout the manuscript.

HF: Corrected and verified.

Lines 123-134: see my comments above on tripartite efflux assemblies – do members of all six efflux pump (super)families form these tripartite assemblies? Or not?

HF:  The text has been modified.

Lines 132-134: misleading/incorrect! See below.

Regarding the driving force, while many pumps are powered by the proton motive force, some MATE family pumps are sodium antiporters and ABC-type pumps rely on ATP hydrolysis (the clue is in the name!). Regarding the internal workings of pumps, the source you cite (46) is a paper entitled Structural Basis of RND-Type Multidrug Exporters. Clearly this is a specific example and cannot be said to apply to all efflux pumps. In reality, while efflux pumps of different families are likely to share structural and functional similarities, they will not be as functionally similar as is implied here.

HF: Thank you for pointing that out. The information has been corrected and an additional sentence on other driving forces has been included.

Table 1:

  • ‘Multidrug; toxic-compound extrusion (MATE) family’ should be ‘Multi- Antimicrobial Extrusion Protein Family’
  • ‘Proteobacterial antimicrobial compound efflux (PACE)family’
  • ‘Resistance-nodulation-division (RND) family’
  • In the ‘Substrates’ column, the first word in each row is sometimes capitalized and sometimes not. Be consistent. Don’t capitalize names of drugs if they aren’t first in the list
  • Use commas rather than semicolons in the substrate’s column. Don’t have spaces before your semicolons/commas
  • Some text in this column is left justified, some is centrally justified. This makes the table look very messy. Again, be consistent.
  • In the ‘Efflux pump’ column, sometimes the name of the protein is given and sometimes the name of the gene. Be consistent, just do one or the other.
  • serovar Typhimurium’ should not be italicized.
  • ‘’ should not be italicized (multiple instances to correct).
  • Methicillin-resistant’ should not be italicized.
  • SM03’should not be italicized.
  • ‘sp T9’ should not be italicized and needs a full stop after the ‘sp’
  • ‘sp. T21’ should not be italicized
  • Be consistent with your usage of abbreviations – e.g. both ‘EtBr’ and ‘ethidium bromide’ appear in this table. Use the full names of compounds only.
  • Why are the majority of the citations in this table italicized? Citation numbers should not be.

HF: At the request of reviewer #2 we modified Table 2 to present data regarding the efflux pumps and substrates identified in P. aeruginosa only. However, we emphasize that the remaining data from the original table were checked and corrected in the new version.

Line 149: what is meant by ‘elements of interactions’? It isn’t clear. Please rewrite sentence.

HF: The sentence has been rewritten

Line 157: what is meant by ‘ligand’ here (you’ve already mentioned a ‘substrate’)? Not sure this sentence makes sense.

HF: The sentence has been rewritten

Line 171: delete ‘present in’

HF: Corrected

Line 172: ‘fulfil’

HF: Corrected

Lines 176-178: wrong tense used.

A: Corrected

Lines 182-184: I’m not quite sure what you are trying to say here – are you trying to say that efflux pumps in bacteria predate the widespread use of antimicrobials by mankind? Sentence needs to be rewritten to be clearer.

HF: The sentence has been rewritten

Lines 184-186: wrong tense used.

HF: Corrected

Line 187: change ‘for’ to ‘from’

HF: Corrected

Line 188: change ‘coding genes’ to ‘genes coding’

HF: Corrected

Line 191: what is meant by ‘its’ here? Sentence needs to rewritten to be clearer.

HF: The sentence has been rewritten

Line 198: removed speech marks

HF: The marks has been removed

Line 200: ‘The presence of efflux pump genes can be considered...’

HF: Corrected

Line 201: where is the citation for this sentence?

A: Thanks for the note. We have included the reference below:

Schmalstieg AM, Srivastava S, Belkaya S, Deshpande D, Meek C, Leff R, van Oers NS, Gumbo T. The antibiotic resistance arrow of time: efflux pump induction is a general first step in the evolution of mycobacterial drug resistance. Antimicrob Agents Chemother. 2012 Sep;56(9):4806-15. doi: 10.1128/AAC.05546-11. Epub 2012 Jul 2. PMID: 22751536; PMCID: PMC3421847.

Line 204: delete ‘the’

HF: Corrected

Lines 209-211: Need to make it clear that extrusion across the periplasm and outer membrane only occurs as part of a tripartite assembly

HF: Corrected, thank you!

Line 214: delete hyphen

HF: Corrected

Line 229: ‘Efflux Pumps’ should not be capitalized

HF: Corrected

Line 243: what do you mean by ‘around the system’? Rewrite to be clearer

HF: The sentence has been rewritten

Line 251: change semicolon to comma

HF: Corrected

Line 257: change ‘the’ to ‘a’

HF: Corrected

Line 304: change semicolon to comma

HF: Corrected

Line 357: ‘et al.’ should be italicized

HF: Corrected

Line 364: ‘et al.’ should be italicized. Also use the abbreviation P. aeruginosa here, not the full name

HF: Corrected

Line 366: should ‘nalB’ and ‘nfxB’ be italicized here? delete ‘formation’

HF: Corrected

Lines 371-373: rewrite sentence to avoid the use of the personal pronoun ‘we’

HF: The sentence has been rewritten

Line 379: ‘in vitro’ should be italicized

HF: Corrected

Line 382: delete comma in ‘carbonyl cyanide, m-chlorophenylhydrazone’

HF: Corrected

Lines 383-384: ‘a class of molecules that has been studied for their efflux pump inhibitory properties’

HF: Corrected

Line 384: delete ‘(a dipeptide)’

HF: Deleted

Line 385: ‘lowering their minimum’

HF: Corrected

Lines 385-386: when you mention MICs here, are you referring specifically to P. aeruginosa? This needs to be clarified since, based on the title of the article, it is definitely implied.

HF: Yes, the information refers to P. aeruginosa. We include this information in the text to make it clearer.

Line 386: what do you mean by ‘same result’ here? If the authors reported it to be less toxic than PAβN, the results can’t have been identical in all assays!

HF: The sentence has been rewritten to improve clarification.

Lines 393-394: sentence makes no sense – are you saying that mefloquine is an existing safe clinical drug or a new molecule? Poor sentence structure, please revise

HF: The aim is to address drug repositioning. Mefloquine is a well-known drug used to treat malaria. In the aforementioned work, the authors demonstrated that mefloquine is also an inhibitor of efflux pumps in P. aeruginosa. The sentence was rewritten to make this information clear.

Figueroa-Romero A, Pons-Duran C, Gonzalez R. Drugs for Intermittent Preventive Treatment of Malaria in Pregnancy: Current Knowledge and Way Forward. Trop Med Infect Dis. 2022 Jul 28;7(8):152. doi: 10.3390/tropicalmed7080152. PMID: 36006244; PMCID: PMC9416188.

Line 394: ‘...which in vitro studies have highlighted...’

HF: Corrected

Lines 397-399: what do you mean here? Assuming we are talking about G. mellonella survival rates, what is the control here? (i.e. sertraline + trimethoprim combination vs. what?) Sertraline has been investigated before as an EPI, could this not actually be evidence of sertraline acting as an EPI rather than trimethoprim?

HF: Sorry for the incomplete information. We rewrite the sentence. In the aforementioned work, the authors found that animals infected with P. aeruginosa treated with the combination of levofloxacin + trimethopim or levofloxacin + sertraline had a longer survival compared to animals treated with the drugs separately.

Lines 398-399: also, you say ‘the effectiveness of both trimethoprim and sertraline in decreasing the overexpression of efflux pumps’. This sounds like you are saying these drugs somehow prevent the expression of the efflux pump proteins – is this what you meant to say? Is there any evidence for this? Are they not actually just acting as traditional EPIs to block important efflux transporters?

HF: The sentence has been rewritten

Line 411: what do you mean by ‘inputs’ here? Consider changing to a different word.

HF: Sorry for the mistake. We refer to the molecules used in the formation of the biofilm. The text has been modified to improve understanding.

Line 529: author list for reference 46 includes a mistake, please correct

HF: Corrected

Reviewer 2 Report

In this paper, Lorusso et al. aimed to review the role of efflux pumps in antimicrobial resistance in P. aeruginosa. However, the role of efflux pumps in AMR has been reviewed extensively in several recent papers:

Manrique et al., 2022 (https://doi.org/10.1111/nyas.14921)

Alav et al. 2021 (https://pubs.acs.org/doi/10.1021/acs.chemrev.1c00055)

Zwama and Nishino, 2021 (https://doi.org/10.3390/antibiotics10070774)

Nishino et al., 2021 (https://doi.org/10.3389%2Ffmicb.2021.737288)

Li & Plésiat, 2016 (https://doi.org/10.1007/978-3-319-39658-3_14)

In particular, the article by Li & Plésiat is a comprehensive review of all reported P. aeruginosa RND efflux pumps as well as pumps from other families. Whereas this review only discusses the role of P. aeruginosa RND pumps, without any information on other families. Therefore, I was not convinced as to whether another review on P. aeruginosa RND pumps is necessary. This review is simply an incremental update on existing P. aeruginosa RND pumps and omits several recent papers as mentioned below in my comments.

Major concerns:

- Section 2 could delve deeper into the context of P. aeruginosa in clinical settings. Currently, it only briefly mentions the clinical significance of P. aeruginosa and omits the problem of P. aeruginosa biofilm infections in cystic fibrosis and chronic wounds. This is required as the authors then later discuss the role of efflux pumps in biofilm formation. 

- As this review article focuses on P. aeruginosa, I did not understand why the authors produced Table 1 listing examples of efflux pumps from other species? Specifically, the ‘Efflux pump’ column is inconsistent and uses a combination of gene names and protein names. The ‘Substrates’ column for some of the efflux pumps is missing a lot of the reported substrates. For instance, the AcrAB-TolC pump exports more than oxacillin, linezolid, novobiocin, and fusidic acid. It can also efflux clinically relevant antibiotics, such as fluoroquinolones, chloramphenicol, and tetracycline. Some substrates were capitalised whereas others were in lower case. Table 1 also lacks the definition of abbreviations for EtBr and TPP. AcrAB-TolC of E. coli is repeated twice, once at the bottom of page 5 and then at the top of page 6. The table omits several P. aeruginosa pumps that have been shown to be able to efflux antibiotics/biocides: 

TriABC-OpmH (https://doi.org/10.1128/JB.00850-07),

MexVW-OprM (https://doi.org/10.1093/jac/dkg390), 

MexPQ-OpmE (https://doi.org/10.1111/j.1348-0421.2005.tb03696.x

I would rather have a table focused on all reported efflux pumps in P. aeruginosa as this would suit the title of the review. For example, the authors could include all reported MATE, MFS, SMR, ABC and PACE pumps of P. aeruginosa.

- Section 6 was very brief and superficial. There have been multiple studies on novel efflux inhibitors of P. aeruginosa pumps that the authors have omitted. Some examples are below:

Tambat et al. (https://pubs.acs.org/doi/10.1021/acsinfecdis.1c00281)

Yamasaki et al. (https://doi.org/10.1128/aac.00672-22)

Mitchell et al. (https://doi.org/10.1128/AAC.00730-1)

Recently, Yamasaki et al. (https://doi.org/10.1128/aac.00672-22) characterised which residues are important for efflux inhibitor binding to MexB. Such studies could be discussed to highlight which residues/binding pockets of these efflux pumps to target with novel compounds.

Minor comments:

- Line 14: There should be an ‘and’ between old and silent.

- Line 21: This statement only applies to tripartite efflux pumps. Therefore, this sentence should be reworded.

- Line 22: Did the authors mean periplasmic protein instead of ‘cytoplasmic protein’?

- Line 23: Pseudomonas aeruginosa should be italicised.

- Line 24: Correct OpRJ to OprJ.

- Line 24-25: Not necessarily true. A total of at least 12 tripartite efflux pumps have been described in Pseudomonas aeruginosa. However, not all of them have clinical relevance if that’s what the authors meant.

- Line 33: Penicillin was not the first antibiotic to be discovered. The first modern antibacterial agent arsphenamine was discovered in 1909 by Paul Ehrlich and his team, which was found to be an effective treatment for syphilis.

- Line 46: The authors should perhaps mention why pharmaceutical companies have moved away from developing novel antibiotics? And what is being done to address the gap in the antibiotic development pipeline.

- Line 62: ‘bacteria’ should not be italicised.

- Line 78: P. aeruginosa should be italicised.

- Line 78-79: The authors should define what MDR and XDR means.

- Line 110: Needs a reference at the end of this sentence.

- Line 125-129: These two sentences are repetitive. 

- Line 204: Reference 134 does not seem to support the intended statement.

- Line 212: The term MFP is not as accurate as PAP (periplasmic adaptor protein).

- Line 223-226: While this is mostly true, there are instances of other pumps that have been reported to be overexpressed in clinical Pseudomonas aeruginosa isolates. For example, mexVW-overexpressing non-cystic fibrosis Pseudomonas aeruginosa isolates have been reported (https://doi.org/10.1139/cjm-2014-0239)

The authors have omitted the Cabot et al. (https://doi.org/10.1128/AAC.01645-10) paper, which reported overexpression of mexYmexBmexF, and mexD in clinical isolates.

- Section 4.1: The authors should include the article by Beig et al. 

(https://doi.org/10.1016/j.genrep.2020.100744) in this paragraph.

- Section 4.3: The authors should include the article by Gomis-Font et al.(https://doi.org/10.1128/AAC.00089-21) in this section. 

- Section 4.3: The mexCD-oprJ efflux genes and the divergently transcribed nfxB repressor has also been detected on megaplasmids from clinical P. aeruginosa isolates (https://doi.org/10.1038/s41467-020-15081-7). More recently, Dong et al. also reported P. aeruginosa isolates carrying tmexCD-toprJ (https://doi.org/10.1016/S2666-5247(22)00221-X).

- Line 296 and 316: What is meant by ‘normal’ conditions? Do the authors mean under laboratory conditions?

- Line 323: Correct MexS to mexS as it’s a gene.

- Line 330: mexEF-oprN should be italicised.

- Line 379: in vitro should be italicised.

- New antibiotics in development against P. aeruginosa, such as LpxC inhibitors, can also be exported by Mex pumps. For example, Jones et al.  (https://doi.org/10.1128/AAC.01490-19) showed that upregulation of efflux pumps MexAB-OprM, MexCD-OprJ, or MexEF-OprN reduced the susceptibility to CHIR-090. The authors should discuss how the Mex pumps present a challenge in antibiotic development and the need for efflux-resistant/efflux non-susceptible compounds. 

- What are the challenges for the development of efflux inhibitors for P. aeruginosa?

- Figure 1. MexXY has been shown to function with OprA or OprM. So why is there no OMF shown? Furthermore, the direction of drug efflux is incorrect. The RND pumps do not capture substrates from the cytoplasm.

- Section 5: There is no context of what biofilms are and their clinical significance. The authors have omitted a recent paper by Harrington et al. (https://journals.asm.org/doi/full/10.1128/aem.01789-21), who showed that the mexGHI-opmD and mexL genes were downregulated and the nalD gene was upregulated in the ex vivo pig lung model of P. aeruginosa cystic fibrosis lung infection.

Author Response

REVIEWER #2

In this paper, Lorusso et al. aimed to review the role of efflux pumps in antimicrobial resistance in P. aeruginosa. However, the role of efflux pumps in AMR has been reviewed extensively in several recent papers:

Manrique et al., 2022 (https://doi.org/10.1111/nyas.14921)

Alav et al. 2021 (https://pubs.acs.org/doi/10.1021/acs.chemrev.1c00055)

Zwama and Nishino, 2021 (https://doi.org/10.3390/antibiotics10070774)

Nishino et al., 2021 (https://doi.org/10.3389%2Ffmicb.2021.737288)

Li & Plésiat, 2016 (https://doi.org/10.1007/978-3-319-39658-3_14)

In particular, the article by Li & Plésiat is a comprehensive review of all reported P. aeruginosa RND efflux pumps as well as pumps from other families. Whereas this review only discusses the role of P. aeruginosa RND pumps, without any information on other families. Therefore, I was not convinced as to whether another review on P. aeruginosa RND pumps is necessary. This review is simply an incremental update on existing P. aeruginosa RND pumps and omits several recent papers as mentioned below in my comments.

HF: Thank you for suggesting the review papers. They are excellent works! As it is a review paper, it is plausible that there is some overlap with other reviews that addressed the same subject. However, the paper needs to be self-sufficient and bring the necessary information to its understanding. From my point of view, all these works are complementary, covering with more details subjects that are more familiar to the authors. For example, none of the reviews addressed the issue of "efflux pumps as a drug target" in depth. However, in our work, there is a section dedicated to the theme. Thus, our aim, as well put by the reviewer, was to incorporate more recent discussions to make an overview of different topics involving AMR, P. aeruginosa and efflux pumps. 

Major concerns:

- Section 2 could delve deeper into the context of P. aeruginosa in clinical settings. Currently, it only briefly mentions the clinical significance of P. aeruginosa and omits the problem of P. aeruginosa biofilm infections in cystic fibrosis and chronic wounds. This is required as the authors then later discuss the role of efflux pumps in biofilm formation.

HF: We appreciate the suggestion. We have included 2 sentences at the end of the first paragraph of section 2 addressing the issue.

- As this review article focuses on P. aeruginosa, I did not understand why the authors produced Table 1 listing examples of efflux pumps from other species? Specifically, the ‘Efflux pump’ column is inconsistent and uses a combination of gene names and protein names. The ‘Substrates’ column for some of the efflux pumps is missing a lot of the reported substrates. For instance, the AcrAB-TolC pump exports more than oxacillin, linezolid, novobiocin, and fusidic acid. It can also efflux clinically relevant antibiotics, such as fluoroquinolones, chloramphenicol, and tetracycline. Some substrates were capitalized whereas others were in lower case. Table 1 also lacks the definition of abbreviations for EtBr and TPP. AcrAB-TolC of E. coli is repeated twice, once at the bottom of page 5 and then at the top of page 6. The table omits several P. aeruginosa pumps that have been shown to be able to efflux antibiotics/biocides: 

TriABC-OpmH (https://doi.org/10.1128/JB.00850-07),

MexVW-OprM (https://doi.org/10.1093/jac/dkg390), 

MexPQ-OpmE (https://doi.org/10.1111/j.1348-0421.2005.tb03696.x) 

I would rather have a table focused on all reported efflux pumps in P. aeruginosa as this would suit the title of the review. For example, the authors could include all reported MATE, MFS, SMR, ABC and PACE pumps of P. aeruginosa.

HF: Sorry for the confusing version of Table 1. We have produced a new version of Table 1 based on reviewer comments.

- Section 6 was very brief and superficial. There have been multiple studies on novel efflux inhibitors of P. aeruginosa pumps that the authors have omitted. Some examples are below:

Tambat et al (https://pubs.acs.org/doi/10.1021/acsinfecdis.1c00281)

Yamasaki et al (https://doi.org/10.1128/aac.00672-22)

Mitchell et al. (https://doi.org/10.1128/AAC.00730-1)

Recently, Yamasaki et al. (https://doi.org/10.1128/aac.00672-22) characterized which residues are important for efflux inhibitor binding to MexB. Such studies could be discussed to highlight which residues/binding pockets of these efflux pumps to target with novel compounds.

HF: Thanks for the suggestion. Suggested references have been included and discussed in section 6.

Minor comments:

- Line 14: There should be an ‘and’ between old and silent.

HF: Thanks for the suggestion. The text has been corrected accordingly

- Line 21: This statement only applies to tripartite efflux pumps. Therefore, this sentence should be reworded.

HF: Corrected

- Line 22: Did the authors mean periplasmic protein instead of ‘cytoplasmic protein’?

HF: Yes, thank you. Corrected.

- Line 23: Pseudomonas aeruginosa should be italicized.

HF: Corrected

- Line 24: Correct OpRJ to OprJ.

HF: Corrected

- Line 24-25: Not necessarily true. A total of at least 12 tripartite efflux pumps have been described in Pseudomonas aeruginosa. However, not all of them have clinical relevance if that’s what the authors meant.

HF: The sentence has been rewritten, thank you!

- Line 33: Penicillin was not the first antibiotic to be discovered. The first modern antibacterial agent arsphenamine was discovered in 1909 by Paul Ehrlich and his team, which was found to be an effective treatment for syphilis.

HF: We apologize for this inconsistency. We added a sentence evidencing Ehrlich's work

- Line 46: The authors should perhaps mention why pharmaceutical companies have moved away from developing novel antibiotics? And what is being done to address the gap in the antibiotic development pipeline.

HF: Thanks for the suggestion. We added the information in the introduction and concluding remarks

- Line 62: ‘bacteria’ should not be italicized.

HF: Corrected

- Line 78: P. aeruginosa should be italicized.

HF: Corrected

- Line 78-79: The authors should define what MDR and XDR means.

A: Corrected

- Line 110: Needs a reference at the end of this sentence.

HF: The reference was added.

- Line 125-129: These two sentences are repetitive. 

HF: The sentence has been rewritten, thank you.

- Line 204: Reference 134 does not seem to support the intended statement.

HF: Thanks for pointing out this inconsistency. Reference 134 has been replaced by the reference below:

Murray JL, Kwon T, Marcotte EM, Whiteley M. Intrinsic Antimicrobial Resistance Determinants in the Superbug Pseudomonas aeruginosa. mBio. 2015 Oct 27;6(6):e01603-15. doi: 10.1128/mBio.01603-15. PMID: 26507235; PMCID: PMC4626858.

- Line 212: The term MFP is not as accurate as PAP (periplasmic adaptor protein).

A: We added it to the text, thank you

- Line 223-226: While this is mostly true, there are instances of other pumps that have been reported to be overexpressed in clinical Pseudomonas aeruginosa isolates. For example, mexVW-overexpressing non-cystic fibrosis Pseudomonas aeruginosa isolates have been reported (https://doi.org/10.1139/cjm-2014-0239)

The authors have omitted the Cabot et al. (https://doi.org/10.1128/AAC.01645-10) paper, which reported overexpression of mexYmexBmexF, and mexD in clinical isolates.

HF: Thanks for the suggestion. We have included reference Cabot et al. along with reference 134 to reinforce the information about overexpression of efflux pumps.

As for the MexVW pump, we recognize its importance. Our decision not to include MexVW in this study was due precisely to the fact that MexVW is not so common. We chose to focus on the four main pumps present in P. aeruginosa.

- Section 4.1: The authors should include the article by Beig et al. (https://doi.org/10.1016/j.genrep.2020.100744) in this paragraph.

HF: Added, thank you

- Section 4.3: The authors should include the article by Gomis-Font et al.(https://doi.org/10.1128/AAC.00089-21) in this section.

A: It is already included in this section (ref. 181 in the original version, 162 post-review)

- Section 4.3: The mexCD-oprJ efflux genes and the divergently transcribed nfxB repressor has also been detected on megaplasmids from clinical P. aeruginosa isolates (https://doi.org/10.1038/s41467-020-15081-7). More recently, Dong et al. also reported P. aeruginosa isolates carrying tmexCD-toprJ (https://doi.org/10.1016/S2666-5247(22)00221-X).

HF: Thank you! We’ve added it to the section.

- Line 296 and 316: What is meant by ‘normal’ conditions? Do the authors mean under laboratory conditions?

HF: Rewritten for clarification. Thanks!

- Line 323: Correct MexS to mexS as it’s a gene.

HF: Corrected

- Line 330: mexEF-oprN should be italicized.

HF: Corrected

- Line 379: in vitro should be italicized.

HF: Corrected

- New antibiotics in development against P. aeruginosa, such as LpxC inhibitors, can also be exported by Mex pumps. For example, Jones et al. (https://doi.org/10.1128/AAC.01490-19) showed that upregulation of efflux pumps MexAB-OprM, MexCD-OprJ, or MexEF-OprN reduced the susceptibility to CHIR-090. The authors should discuss how the Mex pumps present a challenge in antibiotic development and the need for efflux-resistant/efflux non-susceptible compounds.

HF: We’ve added it to the text, thank you!

- What are the challenges for the development of efflux inhibitors for P. aeruginosa?

HF: We appreciate the suggestion. A discussion of the challenges in developing efflux pump inhibitors in P. aeruginosa has been added at the end of section 5.

- Figure 1. MexXY has been shown to function with OprA or OprM. So why is there no OMF shown? Furthermore, the direction of drug efflux is incorrect. The RND pumps do not capture substrates from the cytoplasm.

HF: We’ve corrected and added OprM to the MexXY representation, since OprA is only present in certain strains. We based our representation on the transenvelope efflux model, which has little support, when we should have used the periplasmic efflux model. It’s corrected now, thank you very much!

- Section 5: There is no context of what biofilms are and their clinical significance. The authors have omitted a recent paper by Harrington et al. (https://journals.asm.org/doi/full/10.1128/aem.01789-21), who showed that the mexGHI-opmD and mexL genes were downregulated and the nalD gene was upregulated in the ex vivo pig lung model of P. aeruginosa cystic fibrosis lung infection.

HF: Thanks for the suggestion. We include a brief description of biofilm at the beginning of section 5 and cite the study by Harrington et al at the end of section 5.

Round 2

Reviewer 2 Report

  • The authors have satisfactorily addressed all of my comments/feedback. The review is much more focused and detailed. I applaud the authors for their effort. 

  •  

    Minor comments:

  •  

    Line 84: The authors could cite a seminal review article on the molecular mechanisms that was published recently (https://doi.org/10.1038/s41579-022-00820-y).

  •  

  •  

    Page 12: Figure 1 seems to be replicated twice. I'm assuming the figure in page 12 is the old version.

Author Response

REWIER #2

Comments to the authors

The authors have satisfactorily addressed all of my comments/feedback. The review is much more focused and detailed. I applaud the authors for their effort.

HF: I would like to thank you deeply for your work in reviewing our paper. An unpaid work, done in good faith for the good of Science.

Minor comments:

Line 84: The authors could cite a seminal review article on the molecular mechanisms that was published recently (https://doi.org/10.1038/s41579-022-00820-y).

HF: Thank you for the suggestion. We include the work of Dr. Blair’s group as reference [14].

Page 12: Figure 1 seems to be replicated twice. I'm assuming the figure in page 12 is the old version.

HF: Thank you for the note. Yes, the figure on page 12 is the old one and the figure on page 13 is the current one.
